# POOR TEACHING: EXPLORE AND QUESTION KNOWLEDGE DISTILLATION UNDER DISTRIBUTION SHIFT

## ABSTRACT

Knowledge distillation techniques transfer knowledge from a complex or large learning model into a small model, and have made remarkable achievements in recent decades. However, few studies has investigated and explored the mechanism of the knowledge distillation against distribution shifts in real scenarios. In this paper, we reconsider the knowledge distillation paradigm under the shift situations, by reformulating the objectives of distillation with multiple domains. Under the novel paradigm, we propose a unified and systematic evaluation framework to benchmark knowledge distillation against two general distributional shifts including diversity and correlation shift. The evaluation benchmark covers more than 20 methods from algorithmic, data-driven, and optimization perspectives for five benchmark datasets. Extensive experiments are constructed and some constructive findings are summarized to explain when and how the existing knowledge distillation methods work against distribution shifts.

## 1 INTRODUCTION

Knowledge Distillation (KD) is widely used to transfer knowledge from large model to small model (Gou et al., 2021), with the underlying independent and identically distribution (*i.i.d.*) assumption. However, the efficacy of KD can be limited by distribution shift (non- *i.i.d.* case). Distribution shift refers to a situation where the data distribution during inference differs significantly from that during training (Wiles et al., 2021; Yang et al., 2023).This may lead to a drop in the performance of student model after distillation. For example, while a lightweight student learns to identify dogs or colored digits with the help of the large model, it may not generalize well to real-world samples that it has not seen before, such as cartoon-style dogs or color-reversed digits (Fig. 1). Such shift leads to a trade-off between model capacity and generalization performance on student model (Wang et al., 2021).

As widely applied as KD is, many efforts investigate the mechanics of how knowledge distillation works, and the generalization ability existed in teacher and student model under *i.i.d.* assumption (Allen-Zhu & Li, 2020; Stanton et al., 2021; Hao et al., 2023). However, few have been known or evaluated about when and how well the student model learns from the teacher against distribution shift. This naturally prompt the following research questions:

> Our empirical questions:
>
> *Do existing methods designed for knowledge distillation still work when evaluated under distribution shift? –Especially how well does student models match their teachers across different types of knowledge against distribution shift? What are the optimal data strategies, such as augmentation or pruning, to address distribution shift? Finally, what are the better optimization options in this context?*

To answer the those questions, we propose a knowledge distillation paradigm under distributional shift situations. This novel paradigm involves rethinking the previous knowledge distillation settings and reformulating the objectives to accommodate multiple domains in real-world scenarios. On behalf of this new paradigm, we identify and reconsider the potentially influential factors from three different perspectives: algorithm-level, data-level, and optimization-level. We also specify two

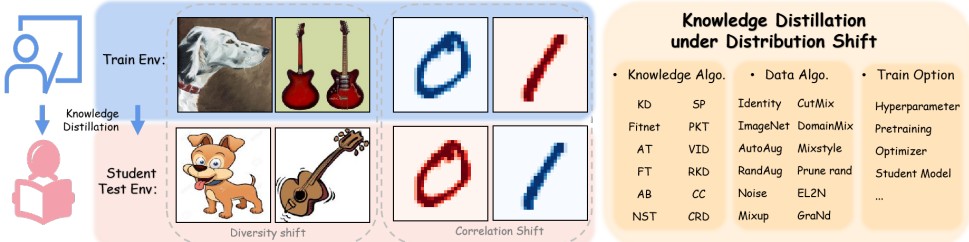

Figure 1: *(Left)* Knowledge distillation tends to exhibit overfitting to distribution shift in real-world applications. Distribution changes are typically categorised as diversity and correlation shift according to the changes in $P(X)$ and $P(Y|X)$, respectively, such as style-changed pictures or color-reversed digits. *(Right)* We propose a unified framework to evaluate knowledge distillation under distribution shift and compare its effectiveness against more than 20 algorithms. The perspective of various algorithms mainly from three level, including algorithm, data, and optimization.

types of distribution shifts with respect to the characteristics of domain data, as previously described by Ye et al. (2022). Furthermore, we propose a comprehensive and systematic evaluation framework to benchmark the effectiveness of knowledge distillation under distributional shift scenarios. The evaluation benchmark covers a wide spectrum of knowledge distillation approaches (more than 20) categorized, consisting of knowledge types, distillation data manipulation, and optimization options.

We conduct extensive experiments in our benchmark, and show that this framework analyzes knowledge distillation performance in various real-world settings, which is truly crucial. It can provide insight into different ways and enhance KD interpretability and stability by regulating the negativity. Besides, our framework is easily extendable and brings new dimensions to the recent emerging areas (*e.g.*, knowledge transfer from foundation models to small models in specific domains). We finally summarize some constructive meta recommendations as takeaways to benefit the research community. Specifically, the main contributions are summarized as follows:

- We formulate a novel knowledge distillation framework evaluated under distribution shift by rethinking previous settings. Our framework considers influential factors and offers a comprehensive insight for tackling such problem.

- We construct a thorough and systematic evaluation benchmark for knowledge distillation under distributional shift situations, which spans a wide spectrum of approaches with more than 20 algorithms. We believe that this is the first work to evaluate the performance of various distillation methods on shifted distribution.

- Based on extensive experiments on our benchmark, we have some constructive findings to explain when and how the existing knowledge distillation methods work against distribution shift. Few knowledge distillation methods can work against all shift conditions, and vanilla knowledge distillation method is enough in some cases. Data manipulation and pre-training can be effective mechanisms to improve the robustness of the student model against distribution shifts.

## 2 FRAMEWORK TO EVALUATE DISTILLATION TO DISTRIBUTION SHIFT

### 2.1 PRELIMINARIES

**Knowledge Distillation (KD).** Hinton et al. (2015) first proposed KD to transfer knowledge from elaborate complex network to a shallower one. Under the supervised classification setting, given a large-scale teacher model $T = T(x; \theta_t)$ optimized on training data $D = \{(x, y)|x \in \mathcal{X}, y \in \mathcal{Y}\}$, where $\mathcal{X}$ and $\mathcal{Y}$ denotes the input and output space, respectively. The lightweight student model $S = S(x; \theta_s)$ directly minimizes the following objective:

$$\min_{\theta_s} \mathbb{E}[\alpha \ell_{KL}(S, T) + (1 - \alpha)\ell_{CE}(S, y)] \tag{1}$$

where the distillation weight $\alpha \in [0, 1]$. $\ell_{KL}$ is the transfer loss term that encourages $S$ to imitate the predictive distribution of $T$, and $\ell_{CE}$ is the cross-entropy between student outputs and ground-

truth labels. In general, equation 1 works under the *i.i.d.* assumption which does not hold in real scenarios, as Fig 1. In the case of $P_{tr}(x, y) \neq P_{te}(x, y)$, the student model might encounters failure, eventually not being applicable.

**KD under distribution shift (non-*i.i.d* case).** In the context of non-*i.i.d* case, we are given $K$ similar but distinct training domains $D_{tr} = \{D^e = (x^e, y^e)\}_{e=1}^{K}$, each are sampled from the joint distribution $P_{\mathcal{X} \times \mathcal{Y}}^e$. Noted that $P_{\mathcal{X} \times \mathcal{Y}}^e \neq P_{\mathcal{X} \times \mathcal{Y}}^{e'}, e \neq e'$ and $e, e' \in \{1, \ldots, K\}$. The goal of KD under distribution shift is to construct a student model $S(x; \theta_s)$ which can perform well on unseen environment $D_{te}$. That is, $D_{te}$ is not accessible in training and $P_{\mathcal{X} \times \mathcal{Y}}^{tr} \neq P_{\mathcal{X} \times \mathcal{Y}}^{te}$. The objective function can be reformulate as:

$$\min_{\theta_s} \mathbb{E}_{(x,y) \in P_{\mathcal{X} \times \mathcal{Y}}^{te}} [\alpha \ell_{KL}(S, T) + (1 - \alpha)\ell_{CE}(S, y)] \tag{2}$$

We focus on offline distillation here, which involves the transfer from a pre-trained teacher model to a student. In theory, we expect the student model can generalize to different distributions that are invisible shifts. However, in practice, the trade-off between handling shifts and model capacity limits the student model. Motivated by this gap between theory and reality, the research question is prompt: *There are many existing methods designed for knowledge distillation, do they still work under distribution shift?* To investigate the question, we propose a evaluation framework of KD against distribution shift. As a whole, our framework try to provide novel insights for previous KD and enhance its interpretability and stability.

## 2.2 DISTILLATION FRAMEWORK UNDER DISTRIBUTION SHIFT

In the proposed evaluation framework, we investigate the effects of distribution shift on distillation process, delving into distinct and complementary viewpoints in real cases, respectively KD algorithms *(algorithm-level)*, data manipulation mechanisms *(data-level)*, and optimization selection *(optimization-level)*. Furthermore, we introduce the precise settings of three specific circumstances that arise under distribution shifts.

**Transferable Knowledge algorithms.** Current KD algorithms can be classified based on the types of transferred knowledge (Gou et al., 2021). While considering the different knowledge used in diverse algorithms, one can reformulate the goal as:

$$\min_{\theta_s} \mathbb{E}_{(x,y) \in P_{\mathcal{X} \times \mathcal{Y}}^{te}} [\alpha \ell_{KL}(S, T) + (1 - \alpha)\ell_{CE}(S, y) + \beta \ell_{reg}] \tag{3}$$

where $\ell_{reg}$ denotes different knowledge sources adopted as the regularization terms of KD and $\beta$ is the trade-off hyperparameter. Based on equation 3, our aim is to explore the trends and reasons for the impact of different knowledge types under distribution shift. Specifically, we seek to answer the question: *What kinds of knowledge can help the student match the teacher model well against distribution shift?*

Thus, we analyze KD algorithms by categorizing them into three distinct categories, as proposed by Gou et al. (2021). **(1) Logit-based knowledge**, which is most popular and imitates the teacher's fully connected layer directly (*vanilla* KD, Hinton et al. (2015)). **(2) Feature-based knowledge**, which refers to student works with specific intermediate layers in its teacher. The different concerns to the hint layers affect feature selection during transferring, such as attention (AT, Zagoruyko & Komodakis (2016)) or neuron (NST, Huang & Wang (2017)). **(3) Relation-based knowledge**, which leverages the relevance of model layers or samples as learned information to guide student's learning, such as contrastive learning (CRD, Tian et al. (2019)) or similarity matrix(SP, Tung & Mori (2019)). Knowledge of the relevance is informative and can be shared to guide learning.

**Distillation Data Manipulation.** The impact of distribution shift is not solely determined by distillation algorithms but also significantly influenced by differences in data. In real-world scenarios, the teacher model trained on insufficient or inadequate data is often imperfect, and the performance of the student model is highly dependent on access to a large and high-quality training dataset. However, in the presence of distribution shift, acquiring such a dataset becomes an insurmountable challenge. It is natural to ask *how do we choose a proper data strategy to ensure the robustness of distillation to distribution shift?* For the situation of data manipulation-based KD, We reformulate the learning objective as:

$$\min_{\theta_s} \mathbb{E}_{(x,y) \in P_{\mathcal{X} \times \mathcal{Y}}^{te}} [\alpha l_{KL}(S, T) + (1 - \alpha)l_{CE}(S^e, y)], (\hat{x}, y) \in P_{\mathcal{X} \times \mathcal{Y}}^{tr} \tag{4}$$

By changing the input $x$ to $\hat{x}$ with manipulation function $M(\cdot)$, this technique can assist students in better learning.

In our framework, we study two manipulation mechanisms of distillation data, including *data augmentation* and *data pruning*: (1) If distillation data is the main cause of poor teaching performance, **data augmentation**, which is a simple and effective way to improve the coverage of data distribution and inter-domain robustness, *i.e.*, $M(x) = \hat{x}$ and $(\hat{x}, y) \in P^{tr}_{\mathcal{X} \times \mathcal{Y}}$. We pay more attention to the augmentation approach based on randomization or generation. Random-based augmentation is typically achieved by creating new complex environments based on randomized manipulation (*e.g.*, AutoAugment (Cubuk et al., 2018), RandAugment (Cubuk et al., 2020)). Additionally, generation-based augmentation concerns the creation of more diverse data at the feature level (*e.g.*, Mixup (Zhang et al., 2017)). (2) What if there was no augmentation, and the student was distilled from important or representative samples? **Data pruning** (Sorscher et al., 2022), which is the approach to quantifying individual sample differences by removing low-quality or immaterial samples in the dataset, *i.e.*, $M(D_{tr}) = \{\hat{D}_{tr} = (\hat{x}, y)\}$ and $(\hat{x}, y) \in P^{tr}_{\mathcal{X} \times \mathcal{Y}}$. We expect to improve the training efficiency and robustness to distributional shift by improving the quality of distillation data.

**Optimization option.** Prior studies have shown that different optimization settings can lead to varying performance on student models in KD (Stanton et al., 2021). It is worth noting that our framework for distribution shift differs from previous studies due to the lack of access to the unknown test environment. The learning process of KD involves several factors that inherently affect the training process, such as hyperparameter selection and the teacher-student architecture. To gain a better understanding of the influence of such factors, we conduct empirical observations with all other variables held constant.

**Types of Distribution shift.** To better study the performance of KD under distributional shift, we propose to characterize how features change in the downstream domain. Inspired by the real world, Ye et al. (2022) formalize the distribution shift into two types of $P^e(X)$ and $P^e(Y|X)$, namely *diversity shift* and *correlation shift*. The nontrivial shifts lead to large differences across environments, rendering KD vulnerable to overfitting with different poor teaching.

**Diversity shift** describes the fact that each environment in the dataset represents a diversity of characteristics in domains. For example, learning the images from the domain of art painting, but testing on the cartoon-style samples (such as Fig 1 *Left*). **Correlation shift** is caused by spurious correlations in the data that have received more recent attention. For example, the MNIST variant, CMNIST (Arjovsky et al., 2019), consists of the digits with red or green, but flipping the strong correlation of colors and labels in different environments (such as Fig 1 *Right*). This thus creates correlation shift that confuses the student. Detailed quantification and examples can be found in Appendix C. By studying these areas, we can better understand poor performance and learn how to deal with it. Further details of the approach will be provided below.

## 3 BENCHMARKING DISTILLATION FRAMEWORK

We present a evaluation framework aimed at explore the various KD methods in response to distributional changes. Our framework involves proposing an benchmark and assessing over 20 algorithms that span a wide spectrum of approaches, including algorithmic, data-based, and optimization-based techniques. Furthermore, we include the classical benchmark datasets from various domains across distribution shits. We believe that our study provides the first systematic evaluation on the effectiveness of various distillation techniques for distribution shift. Specifically, the distillation benchmark is explored by covering the following areas:

**Knowledge Transfer Algorithms.** We first focus on the KD algorithm, and introduce the following transfer methods in the benchmark, which we categorize into three distinct aspects. (1) *Logit-based knowledge*: Knowledge Distillation (**KD**) (Hinton et al., 2015). (2) *Feature-based knowledge*: **Fitnet** (Romero et al., 2014), Attention Transfer (**AT**) (Zagoruyko & Komodakis, 2016), Factor Transfer(**FT**) (Kim et al., 2018), Activation Boundaries (**AB**) (Heo et al., 2019), Neuron Selectivity Transfer (**NST**) (Huang & Wang, 2017).(3) *Relation-based knowledge*: Similarity-Preserving (**SP**) (Tung & Mori, 2019), Probabilistic Knowledge Transfer (**PKT**) (Passalis & Tefas, 2018), Variational Information Distillation (**VID**) (Ahn et al., 2019), Relation Knowledge Distillation (**RKD**) (Park et al., 2019), Correlation Congruence (**CC**) (Peng et al., 2019a), Contrastive Representation Distillation

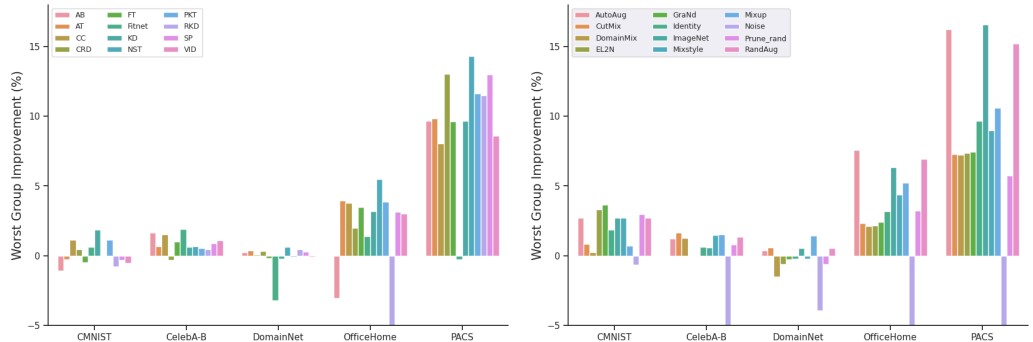

Figure 2: Worst-group improvements over ERM on different shifted datasets. *Left:* data manipulation algorithms; *Right:* Knowledge Transfer algorithms. Different algorithms can achieve enhancement only on certain shift types.

(**CRD**) (Tian et al., 2019). For further details and comprehensive descriptions of the algorithms used, please refer to Appendix D.2.

**Data Manipulation Techniques.** The unseen distribution encourages us to choose suitable manipulation techniques to increase the data quality and diversity. The manipulation algorithms in our benchmark contain the following areas of specific interest: (1) *vanilla*: **Identity**. (2) *Data Augmentation*: a) *Random-based Augmentation*: **ImageNet baseline**, **AutoAugment** (Cubuk et al., 2018), **RandAugment** (Cubuk et al., 2020), **Gaussian noise**, b) *Generation-based Augmentation*: **Mixup** (Zhang et al., 2017), **CutMix** (Yun et al., 2019), **DomainMix** (Wang et al., 2020), **MixStyle** (Zhou et al., 2021). (3) *Data Pruning*: **Random Prune**, **EL2N** (Paul et al., 2021), **GraNd** (Paul et al., 2021). For further details and comprehensive descriptions of the data techniques used, please refer to Appendix D.1.

**Optimization Option.** We also investigate the effectiveness of various optimization options on KD process in adressing distribution shift. Concretely, we examine a range of optimization options, which include **(1) Distillation hyperparameters.** We perform sweep for hyperparameters and set for all, which ensures fair comparisons in different settings. **(2) Pretrain or not.** Students pretraining on ImageNet is a prevalent out-of-the-box solution, which provide more relevant and diverse features for the target domains, and thus improve the performance. In our benchmark, we add the pre-training option, and observe whether is useful. **(3) Optimizer.** We allow two common optimizers Adam (Kingma & Ba, 2014) and SGD (Bottou, 2012) in our evaluation, and study the potential shift effects of different optimizers. **(4) Student Selection.** Learning ability could be specific to student capacity and ResNet-like architectures. In our benchmark, we also replace the student model with another in order to investigate relevance, such as WRNet and MobileNet (ResNet by default).

**Shifted Datasets**. We choose five datasets for evaluation to explore the performance of KD on the two-dimensional shift of diversity and correlation. In terms of divesity shift, we use **PACS** (Li et al., 2017), **OfficeHome** (Venkateswara et al., 2017), and **DomainNet** (Peng et al., 2019b) . Regarding correlation shift, we construct the modified version of MNIST and CelebA (Liu et al., 2015) based on recent work, also known as **CMNIST** (Arjovsky et al., 2019) and **CelebA-Blond** (Ye et al., 2022). For the constructed datasets, the training and test domains are pre-defined and fixed. More quantitative information and description of the datasets are shown in Appendix D.1.

**Evaluation Implementation.** The following implementation choices are highlighted to achieve a consistent and realistic assessment setting. *(1) Model choice.* To better understand the robustness of knowledge distillation under distribution shift, we select **ResNet-50/18** as the base teacher/student model, which is a common choice for previous distillation algorithms. ResNet is pretrained in ImageNet and then distilled for each dataset. *(2) Model selection criterion.* We using the same model selection criterion as the out-of-distribution generalization community, and there is still no consensus recently. We choose **Training-Domain Validation** as a criterion for consistency with existing work (Gulrajani & Lopez-Paz, 2020). *(3) Evaluation Metrics.* In our benchmarks, we include two metrics used as different aspects of the evaluation criteria. Along with the gold standard,

Table 1: Average accuracy for all KD algorithms on datasets dominated by diversity/correlation shift. These experiments compare more than 12 popular algorithms in five benchmarks under identical conditions. $\Delta \uparrow$ denotes the improvement over vanilla KD.

| | Method | PACS | OfficeHome | DomainNet | CelebA-B | CMNIST | Avg. | $\Delta \uparrow$ |
|---|---|---|---|---|---|---|---|---|
| | Teacher | 82.59 | 70.74 | 38.80 | 84.74 | 11.93 | 57.8 | |
| | Student | 75.99 | 63.47 | 34.42 | 83.91 | 11.01 | 53.8 | |
| | KD | $81.12 \pm 0.65$ | $65.44 \pm 0.36$ | $37.50 \pm 0.02$ | $84.55 \pm 0.38$ | $12.86 \pm 1.15$ | 56.3 | - |
| Feature-based | Fitnet | $73.20 \pm 0.90$ | $60.51 \pm 0.80$ | $23.86 \pm 2.47$ | $85.82 \pm 1.63$ | $11.65 \pm 1.15$ | 51.0 | -9.39% |
| | AT | $80.72 \pm 0.49$ | $65.51 \pm 0.13$ | $37.39 \pm 0.07$ | $84.58 \pm 0.08$ | $10.73 \pm 0.66$ | 55.8 | -0.90% |
| | FT | $79.40 \pm 0.32$ | $62.96 \pm 0.30$ | $35.76 \pm 0.03$ | $84.90 \pm 1.12$ | $10.54 \pm 0.27$ | 54.7 | -2.81% |
| | AB | $76.51 \pm 0.48$ | $54.83 \pm 1.10$ | $29.98 \pm 0.21$ | $85.57 \pm 1.22$ | $9.92 \pm 0.15$ | 51.4 | -8.76% |
| | NST | $82.05 \pm 0.32$ | $65.65 \pm 0.29$ | $38.05 \pm 0.06$ | $84.81 \pm 0.18$ | $10.69 \pm 1.35$ | 56.3 | -0.08% |
| Relation-based | SP | $81.59 \pm 0.51$ | $65.20 \pm 0.04$ | $34.67 \pm 0.14$ | $84.58 \pm 2.21$ | $11.05 \pm 0.67$ | 55.4 | -1.56% |
| | PKT | $81.47 \pm 0.58$ | $65.72 \pm 0.17$ | $37.93 \pm 0.09$ | $84.46 \pm 1.83$ | $12.15 \pm 1.29$ | 56.3 | 0.09% |
| | VID | $80.45 \pm 0.91$ | $65.42 \pm 0.20$ | $37.55 \pm 0.12$ | $85.01 \pm 1.45$ | $10.49 \pm 0.32$ | 55.8 | -0.91% |
| | RKD | $76.11 \pm 0.71$ | $46.12 \pm 1.06$ | $35.33 \pm 0.15$ | $84.37 \pm 0.98$ | $10.24 \pm 0.13$ | 50.4 | -10.41% |
| | CC | $80.42 \pm 1.00$ | $65.30 \pm 0.02$ | $37.14 \pm 0.08$ | $85.43 \pm 0.35$ | $12.15 \pm 1.20$ | 56.1 | -0.37% |
| | CRD | $79.66 \pm 0.15$ | $63.91 \pm 0.40$ | $37.75 \pm 0.04$ | $83.59 \pm 1.82$ | $11.48 \pm 1.28$ | 55.3 | -1.80% |

**Average Accuracy**, we also report **Worst Group Accuracy** as evaluation criterion. The average and standard deviation were reported in Section 4 based on three seeds.

## 4 A FINE-GRAINED ANALYSIS

### 4.1 PERFORMANCE ACROSS TRANSFER ALGORITHMS

**KD can help lightweight model to alleviate the shift effect, but not consistently.** As Fig. 2a illustrates, we find that all KD algorithms can be an effective technique for improving worst-group accuracy of the lightweight model, as distillation makes training easier by providing more label information. However, we also observe in Table 1 that none of the KD methods consistently outperforms concerning the average accuracy in both directions of shift. For example, while improvements are available on datasets dominated by diversity shift, none of them can identify well on CMNIST with a strong correlation shift. The teacher learns the spurious correlation of color and label from training data, which reverses in the testing data, finally leads to poor teaching results.

***Vanilla* is better, and more complex algorithms offer limited improvement.** More complex algorithms (vs. *vanilla*) perform well on wosrt-group case as depicted in Fig. 2a. However, upon further zooming in Table 1, most of complex methods may not be as effective, such as AB and RKD. Most of them cannot surpass *vanilla* KD in the most general and complicated cases of average accuracy. This supports our view that current KD algorithms remain largely vulnerable to distribution shift, yet still be effective in supporting worst-case. In addition, we also find that methods that learn from the relevance in teacher models are more generically valid, such as PKT. In contrast to relevance-based, feature-based students are worse on average accuracy. These findings inspired us to further understand the role of knowledge source in KD against distribution shift, and guides us to design new algorithms.

**Negative knowledge may be transferred from hint layer.** Our results demonstrate that the logit-based approach outperforms the others in terms of generalizability to distribution shift. We perform a center kernel analysis (CKA) (Kornblith et al., 2019) to analyze the reason and compare the features of the layer extracted by the teacher and different students. As shown in Fig. 3, the CKA of Fitnet highlights huge diversity of representation within the student model, except for the poor performance. This difference can be attributed to the distinct ability of the teacher and the student to learn from multiple domains. This again confirms previous observations in Table 4 that mimicking the teacher's middle layer directly is not suitable when handling distribution shift, which prevents the alignment process of features.

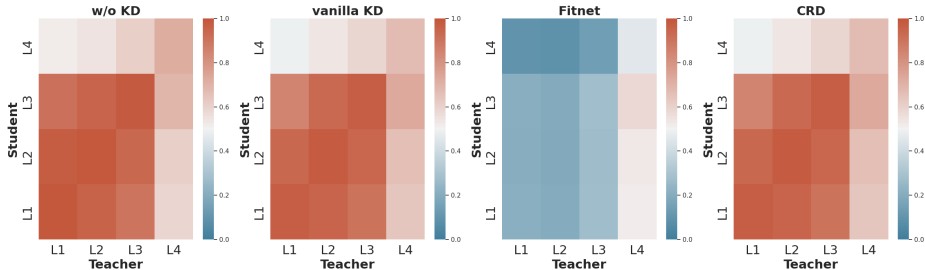

Figure 3: Representation similarity heatmap measured by CKA. L1 ∼ L4 represents the representation extracted from the first to fourth hint layer. The CKA representations inside the student model differs significantly based on their knowledge sources.

Table 2: Average accuracy for all data techniques on datasets dominated by diversity/correlation shift. These experiments compare 12 popular algorithms in five benchmarks under identical conditions, showing the robustness and improvement potential of data techniques. $\Delta \uparrow$ denotes the improvement over vanilla KD.

|  | Method | PACS | OfficeHome | DomainNet | CelebA-B | CMNIST | Avg. | $\Delta \uparrow$ |
|---|---|---|---|---|---|---|---|---|
|  | *vanilla* KD | $81.12 \pm 0.65$ | $65.44 \pm 0.36$ | $37.50 \pm 0.02$ | $84.55 \pm 0.38$ | $12.86 \pm 1.15$ | 56.3 | - |
| Augmentation | + ImageNet | $83.24 \pm 0.10$ | $67.31 \pm 0.22$ | $36.80 \pm 0.07$ | $84.48 \pm 0.20$ | $13.73 \pm 0.60$ | 57.1 | 1.45% |
|  | + AutoAug | $83.56 \pm 0.28$ | $67.44 \pm 0.17$ | $36.73 \pm 0.08$ | $85.15 \pm 0.32$ | $13.73 \pm 0.60$ | 57.3 | 1.83% |
|  | + RandAug | $83.00 \pm 0.61$ | $67.26 \pm 0.30$ | $37.11 \pm 0.03$ | $85.27 \pm 0.44$ | $13.73 \pm 0.60$ | 57.3 | 1.74% |
|  | + Mixup | $80.07 \pm 0.67$ | $65.74 \pm 0.06$ | $39.39 \pm 0.09$ | $85.43 \pm 1.04$ | $11.73 \pm 0.76$ | 56.5 | 0.32% |
|  | + CutMix | $78.70 \pm 0.37$ | $65.07 \pm 0.40$ | $38.00 \pm 0.18$ | $85.54 \pm 0.71$ | $11.83 \pm 1.30$ | 55.8 | -0.83% |
|  | + DomainMix | $79.61 \pm 0.22$ | $65.43 \pm 0.10$ | $34.84 \pm 0.13$ | $85.17 \pm 0.35$ | $11.24 \pm 1.08$ | 55.3 | -1.84% |
|  | + Mixstyle | $80.63 \pm 0.39$ | $65.90 \pm 0.46$ | $37.50 \pm 0.02$ | $85.38 \pm 0.17$ | $13.73 \pm 0.60$ | 56.6 | 0.59% |
|  | + Noise | $66.78 \pm 2.27$ | $52.74 \pm 1.69$ | $18.84 \pm 0.21$ | $76.38 \pm 1.93$ | $10.37 \pm 0.72$ | 45.0 | -20.02% |
| Pruning | + Prune_rand | $79.43 \pm 0.39$ | $64.78 \pm 0.22$ | $37.09 \pm 0.01$ | $84.69 \pm 0.49$ | $13.99 \pm 0.33$ | 56.0 | -0.53% |
|  | + EL2N | $79.02 \pm 1.69$ | $64.20 \pm 0.34$ | $36.24 \pm 0.04$ | $83.89 \pm 1.49$ | $14.33 \pm 1.01$ | 55.5 | -1.35% |
|  | + GraNd | $79.05 \pm 0.54$ | $64.14 \pm 0.04$ | $36.82 \pm 0.02$ | $83.89 \pm 1.91$ | $14.65 \pm 0.12$ | 55.7 | -1.04% |

## 4.2 THE ROLE OF DISTILLATION DATA

**Data augmentation helps, but no forever winner.** As Fig. 2b (right) reveals, the effectiveness of augmentation depends on the realistic situation of datasets. Any form of data augmentation can be effective as it increases the likelihood of a student matching across multiple domains. However, different strategy shows diverse results in Table 2. Of all shift settings, random-based augmentation typically improves performance, while generation-based augmentation performs better on correlation shift, *e.g.*, AutoAugment versus CutMix. It is worth noting that the Gaussian noise falls on all datasets, decreasing significantly compared to our baseline. Noisy information brings incorrect knowledge and aggravates teacher-student disagreement brought by shift.

**Students learning with random-based augmentation perform better.** As Table 2 summarizes, random-based augmentation (*i.e.*, RandAugment and AutoAugment) are combinations of different tricks, their performances are more changeable (with close mean but higher variance) when compared to ImageNet baseline. Generation-based augmentation has shown to be a powerful tool such as Mixup, but vice versa in our benchmark. We also can find DomainMix and CutMix performs poor on the dataset with diversity and correlation shift. Our analysis has led us to the conclusion that in the case of distribution shift, the augmentation effect for KD needs to satisfying two key conditions, one being close to ground-level truth and the other being close to the teacher's knowledge. The above observation is also consistent with (Beyer et al., 2022) in that consistent image view is the key to a sound practice of KD even under distribution shift.

**Data pruning makes sense for KD under distribution shift.** Our observation in Table 2 suggest that not all training samples are equal, with data pruning exploring data quality required for distillation by keeping important examples. Specifically, we observed that selecting 75% of all samples and utilizing distillation techniques yielded a student model that retained over 95% of its pre-

Table 3: The performance of Pretraining and Optimizer Selection on PACS (%). The training strategy can have a strong impact on the distribution shift during the distillation process.

| Method | Art | Cartoon | Photo | Sketch | Avg. |
|---|---|---|---|---|---|
| Baseline | 81.5 | 78.1 | 95.4 | 69.4 | 81.1 |
| w/ Adam | 51.7 | 62.9 | 70.7 | 61.2 | 61.6 |
| w/o Pretrain | 33.6 | 40.6 | 51.3 | 27.4 | 38.2 |

Table 4: Results of different layers on $\ell_{reg}$ (%). L1 $\sim$ L4 represent the distillation using the knowledge from the first to fourth hint layer.

| Method | Art | Cartoon | Photo | Sketch | Avg. |
|---|---|---|---|---|---|
| KD | 81.5 | 78.1 | 95.4 | 69.4 | 81.1 |
| w/ L1 | 74.5 | 75.3 | 91.0 | 62.0 | 75.7 |
| w/ L2 | 70.3 | 72.9 | 89.0 | 54.7 | 71.7 |
| w/ L3 | 67.4 | 71.2 | 88.9 | 61.5 | 72.3 |
| w/ L4 | 81.5 | 78.2 | 94.5 | 74.2 | 82.1 |

pruning performance. Moreover, we found that random pruning outperformed other well-designed metrics. Interestingly, our results indicate that data pruning even can outperform data augmentation techniques in correlation shift-dominated datasets, such as CMNIST. Overall, our study provides empirical evidence highlighting the significance of carefully selecting data for distillation purposes. These insights could inform the creation and utilization of distillation datasets.

### 4.3 POSSIBLE CAUSES ON TRAINING OPTION

In addition to our primary findings, we conducted a thorough examination of key factors that impact the optimization program. Our investigation yielded several intriguing observations, which are as follows. **(1) Pretraining is helpful.** Pretraining on ImageNet can be a valuable technique for distillation tasks to distribution shift, provided that the features learned during pretraining are beneficial for the student's performance (Table 3). **(2) SGD for KD on distribution shift, not Adam.** While Adam and SGD are typically comparable in their performance in KD, our evaluation revealed a notable deviation from this norm. Specifically, we observed a significant decrease in performance when utilizing Adam compared to SGD in Table 3. We speculate that it may be that adaptive optimizers (*e.g.*, Adam) are more prone to overfitting in the face of shift. **(3) Results of other student architectures.** We further evaluate our benchmark using other small networks. KD still has an overall boosting effect on tiny models. However, the extent of gain varies depending on the domain. (Appendix Fig. 6). KD can generally improve the performance of Mobilenet on Art and Cartoon, but not on Photo or Sketch in PACS dataset. These results suggest that improving the robustness of these models to distribution shift may be challenging, particularly in the worst-case scenario.

### 4.4 MORE OBSERVATIONS ON OUR BENCHMARK.

**Performance across different types of shift.** Several algorithms improve much better over the pre-KD period on the situation of diversity shift. However, in the case of CMNIST and CelebA, most of the algorithms can not minimize the gap between teacher and student. Consequently, we argue that previous KD algorithms remain largely vulnerable to spurious correlations. In particular, Empirical Risk Minimization (ERM) still achieves good results, but the problem is that students receive 'bias' in the teacher model while being taught. The use of data manipulation can get rid of bias to some extent, but this is limited by how relevant the manipulated data is to the ground truth.

**High-quality KD data is needed.** The distillation data needed for student model can be original or after augmentation or pruning. This depends on the structure and goals of KD model. Augmenting the data can enhance its diversity and robustness, while pruning can reduce redundancy and noise. However, in any case, the aim of manipulation is to make some useful change to our training data so that we can get more high-quality data and its distribution is closer to the invariant one across any distribution.

**Different students fall into diverse activation maps.** We provide a preliminary visualization from the target domain in Fig. 4 using class activation map (CAM) (Selvaraju et al., 2017). We observe that distillation can help student model to learn more exact and specific invariant representations under distribution shift, e.g., the face of the dog which is focused by learning from the teacher model (Fig. 4b and 4c). We further compare CAM between the data and knowledge transfer algorithms and find that after different forms of knowledge transfer, the key discriminant area of students changes while not much change for data augmentation (Fig. 4d, 4e and 4f).

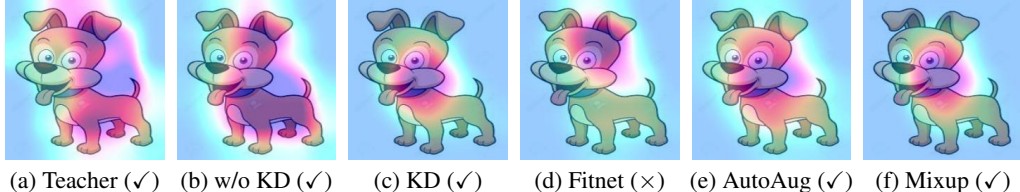

(a) Teacher (✓)    (b) w/o KD (✓)    (c) KD (✓)    (d) Fitnet (×)    (e) AutoAug (✓)    (f) Mixup (✓)

Figure 4: The visualization of Grad-CAM from different models using PACS datset. Models after distillation pay more attention to the region of interest for specific characteristics. ✓/ × denotes whether the prediction is correct or not. KD algorithms can alter the focus of the student, with data techniques the other way round.

**Knowledge from shallow hint layers misleads the student while distribution shift.** To analyze the influence of knowledge in different hint layers, we conduct ablation experiments on four variants of the distillation loss term. As reported in Table 4, we observe an obvious performance drop after matching the first to third layer in the teacher model. Unlike (Romero et al., 2014), which tended to choose the middle layer of the student model as the guiding layer, we find that the fourth layer achieves the best. The upshot is, that knowledge from shallow layers in the teacher model may mislead the student under distribution shift.

## 5   META RECOMMENDATION AND CONCLUSION

Distribution shift resulted in poor KD, so there must be no free lunches in this situation when choosing the KD method. Here are some helpful recommendations to consider according to **various goals of distribution shift**.

**REC 1: Knowledge distillation works.** Neural networks deployed on-device with tiny capacity and low complexity, increasing their susceptibility to overfitting under distribution shift. Therefore, lightweight models should be trained separately from large scale, and KD is much better to achieve this with our benchmarking observation.

**REC 2:** *Vanilla* **is enough, and exploring novel algorithms is required.** Complex transfer patterns offer limited improvement. Feature- and relation-based KD lead to limited improvements across the different types of distribution shift. Of these approaches, vanilla KD generally performs well to distribution shift, and few other complex methods consistently outperform KD by itself. How to make these methods robust to shift is still an open question, especially for the correlation shift.

**REC 3: Using augmentation for robust KD performance** Augmentation closed to the ground-level distribution is more robust to KD for shifted data. If data augmentation can help KD to extract invariant features and extrapolate to unseen environments, please use them. An additional suggestion is that if data augmentation fails to provide well assistance, it presents a promising avenue for further to explore novel methods at data level to enhance the shift robustness of the student model.

**REC 4: Do not forget pretraining and SGD.** Distribution shift affects the choice of training strategy. Pretraining can help distill to more robust features when the relevance of the downstream task is high. And SGD is superior to Adam in knowledge distillation for distribution shift. An area to be studied is the choice of KD training strategy when the shift occurs.

**Conclusion.** We formulate the novel knowledge distillation paradigm under distributional shift situations and broaden the learning objectives of knowledge distillation to multiple domains to address the distribution shifts in real application scenarios. We propose a systematic evaluation framework from three diverse perspectives including the knowledge distillation algorithms, data manipulation mechanisms and optimization options, and we take a comprehensive evaluation benchmark covering more than 20 methods for five benchmark datasets. Several novel insights and tips based on our benchmark are summarized to allow the research community to find optimal solutions when applying knowledge distillation techniques in real scenarios. We also provide further discussion on the limitations and open questions of our benchmark in Appendix B

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
