# OpenReview forum: "Poor Teaching: Explore and Question Knowledge Distillation under Distribution Shift"
_ICLR.cc/2024/Conference — ICLR 2024 Conference Withdrawn Submission_

### Official Review · Reviewer_Uh6z · 2023-10-24

**Soundness:** 2 fair
**Presentation:** 1 poor
**Contribution:** 1 poor
**Rating:** 1
**Confidence:** 5

**Summary:**

The authors of this submission aim to study how student models, trained via various knowledge distillation (KD) approaches, perform under distribution shift, i.e., when KD is performed on different datasets than the target dataset.

Specifically, the authors report results on multiple datasets and study how hyperparameters and design choices (e.g., choice of optimizer, data augmentation) and different KD algorithms affect the students' performance on a given test dataset.

The authors argue that no method consistently outperforms 'vanilla' logit-based KD (Hinton et al.,  2015) and that data augmentation as well as the choice of the the optmizer can help improve performance.

**Strengths:**

**S1**: Better understanding failure cases of current knowledge distillation approaches is an important topic, as the computational gap between models that are run on edge devices and the most performant large models is increasing.

**S2**: The authors evaluate a wide range of different KD approaches and the finding that more complex KD approaches only offer limited gains over 'vanilla' KD is interesting.

**Weaknesses:**

Unfortunately, in the current state this submission does not meet the bar for acceptance at ICLR. Some of the biggest concerns are the following:

**W1 (presentation unclear)**: First and foremost, the presentation and organisation of the manuscript are severely lacking. Even after reading the manuscript several times, the exact setup studied remains unclear to me. Given that the authors aim to introduce a 'benchmark for KD under distribution shift', this is a major concern. To name a few open questions: what data are the students trained on for any given dataset? What data are the teachers trained on? Why should one hope to obtain students that are robust to the colour shift in CMNIST if the teachers are not robust either (see table 1)?

**W2 (insufficient contribution)**: As far as I can follow, the authors seem to evaluate 12 existing KD approaches on 5 different datasets (as discussed in W1, the exact setting remains unclear), apply various augmentation techniques during training, and optimise the hyperparameters for the different approaches. While the breadth of KD approaches and a grid search over the hyperparameters are laudable, this seems to be a minimal requirement for an experimental comparison rather than a sufficient contribution. Of course, a thorough experimental evaluation can be highly valuable if it allows for obtaining new insights or develop novel hypotheses that can be further explored in future work. However, in this submission, the lack of clarity w.r.t. the setting (see W1) make this difficult and the findings discussed by the authors (e.g. data augmentation helps performance) are not sufficiently novel.

**W3 (technical soundness)**: The authors make several statements that are unclear or that lack a thorough experimental validation. Some examples:
   - "[Our benchmark] can provide insight into different ways and enhance KD interpretability and stability by regulating negativity." In which way does the benchmark increase KD interpretability or stability? What do these terms mean in this context? What does "regulating negativity" mean?
   - "In theory, we expect the student model can generalize to different distributions that are invisible shifts." Again, it is unclear to me what this means.
   - "These findings inspired us to further understand the role of knowledge source in KD against distribution shift, and guides us to design new algorithms." It is unclear to me which 'new algorithms' the authors refer to here.
   - "Different students fall into diverse activation maps." + Figure 4: the CAM visualisations of a single image are not sufficient to support the conclusions drawn by the authors.
   - More generally, the lack of clarity with respect to the experimental setup (see W1) makes it difficult to draw robust conclusions from the presented results

**Questions:**

Please see weaknesses.

---

### Official Review · Reviewer_m1kT · 2023-10-31

**Soundness:** 3 good
**Presentation:** 3 good
**Contribution:** 2 fair
**Rating:** 5
**Confidence:** 3

**Summary:**

This work benchmarks the knowledge distillation under the setting of distribution shift, where the training data for teacher network differs from the data for student network ,e.g. style. Two types of distribution shift is studied: diversity shift and correlation shift. The authors also study the effect of data augmentation as a mean to change data distribution. They conduct extensive experiment over 20 publicated KD methods and summarize some suggestion.

**Strengths:**

1. The author systematically investigate the problem of distribution shift for knowledge distillation. The topic is interesting and important for KD in real-world application.
2. The experiment is solid, convering 20 publicated KD method, different data augmentation and benchmarks.
3. The findings in 4.1 and 4.2 is empirical and helpful.
4. The presentation and writing is good.

**Weaknesses:**

1. In eq 2 (page 3), the authors assume that the training data for teacher network is not accessable. This **assumption** makes the problem of distribution shift significant. However, recent work [1] has proposed the data-free KD. They do not need to access the training data for teacher network and still can perform KD. Thus, I think the assumption in eq 2 may be weak.
2. Regarding the type of distribution shfit, the authors study the diversity and correlation shift. I think the author can consider another type. [2] proposed the nasty teacher, where the dark knowledge (representation distribution) of teacher network is *corrupted* and thus is undistillable. [3] lately study how to distill the corrupted teacher knowledge, where the representation distribution of student is normal.
3. In terms of the data augmentation, I think the discussion on section 4.2 has some overlap with [4]. More discussion is welcome.
4. I believe the findings in 4.1 and 4.2 are solid given so much experiments. A concern is that some findings are not consistent on all situation. For instance, the performance on Fig. 2a.

[1] Learning to Learn from APIs: Black-Box Data-Free Meta-Learning, ICML 2023

[2] Undistillable: Making A Nasty Teacher That CANNOT teach students. ICLR 2022

[3] Distilling the Undistillable: Learning from a Nasty Teacher. ECCV 2022

[4] What Makes a “Good” Data Augmentation in Knowledge Distillation – A Statistical Perspective. NIPS 2022

**Questions:**

Please refer to the weaknesses part.

---

### Official Review · Reviewer_ditG · 2023-11-02

**Soundness:** 2 fair
**Presentation:** 3 good
**Contribution:** 2 fair
**Rating:** 3
**Confidence:** 5

**Summary:**

This study re-evaluates knowledge distillation in the context of distribution shifts. A new paradigm adjusts distillation objectives for multiple domains. An evaluation framework is introduced to test knowledge distillation against diversity and correlation shifts, assessing over 20 methods across five datasets. The research offers insights into how current methods handle distribution shifts.

**Strengths:**

The paper delves into an intriguing question: Can methods designed for knowledge distillation maintain their efficacy under distribution shifts? The topic is both timely and relevant.

**Weaknesses:**

There appear to be some gaps in the paper's exploration. A crucial point that seems to be overlooked is the role that teacher models play in Knowledge Distillation (KD). The choice of models, like ResNet50 compared to the CLIP model, can yield drastically different results and insights.

From my own research and experimentation, I've observed that the network architecture has a significant influence on both KD and Domain Generalization (DG). More recent architectures, such as ConNext or VIT variants, might interact differently with KD compared to traditional CNNs.

It wasn't surprising to find that most KD methods falter under distribution shifts. Past research on KD has shown that many methods don't even perform optimally on In-Distribution (I.ID), making it less likely they'd excel in Out-of-Distribution (OoD) scenarios. Additionally, the paper's finding that data augmentation methods are effective for DG was somewhat expected. Given that many DG benchmarks aren't large-scale, it stands to reason that random data augmentations would be beneficial.

Lastly, while the paper offers insights into the performance of Pretraining and Optimizer Selection, the scope seems limited. Drawing conclusions solely from the PACS dataset may be premature. It's a stretch to claim that SGD outperforms Adam based on this limited data. Furthermore, the advantage of using pre-trained weights over non-pretrained ones is a well-established fact in the literature, so it doesn't present a novel revelation in this context.

**Questions:**

See weakness above

---

### Official Review · Reviewer_Ywf2 · 2023-11-08

**Soundness:** 1 poor
**Presentation:** 2 fair
**Contribution:** 2 fair
**Rating:** 3
**Confidence:** 4

**Summary:**

This paper explores knowledge distillation under distribution shifts. It proposes a systematic evaluation framework to benchmark knowledge distillation against diversity and correlation shifts, covering more than 20 methods from algorithmic, data-driven, and optimization perspectives for five benchmark datasets. The paper presents extensive experiments and findings to explain when and how existing knowledge distillation methods work against distribution shifts.

**Strengths:**

1. This paper is well organized. It categorizes the benchmark into KD algorithms, data manipulation techniques, and optimization options.

2. This paper conducts extensive experiments.

3. The logical progression is well-articulated and the writing is easy to follow.

**Weaknesses:**

1. My major concern is that the KD methods do not include recent works. I understand a benchmark paper may select representative methods, and it is not practical to include all the related methods. However, all the compared KD algorithms in this paper are before 2020, which is too old for a 2023 manuscript. There are several recent works that the authors may consider including:
The primary concern highlighted is the omission of recent advancements in Knowledge Distillation (KD) methods in the paper. While it is reasonable for a benchmark study to focus on representative techniques, the absence of any KD algorithms post-2019 in a manuscript for 2023 is conspicuous. To maintain relevance and scholarly rigor, the inclusion of several works from the past few years should be considered. This will not only enhance the comprehensiveness of the paper but also ensure that the evaluation reflects the current state of the art in KD methods. There are several recent works that the authors may consider:
 - Annealing KD (Jafari et al., 2021);
 - DKD (Zhao et al., 2022);
 - FilterKD (Ren et al., 2022);
 - MetaDistill (Zhou et al., 2022);
 - PTLoss (Zhang et al., 2023).

2. The analysis part just provides limited insights. I personally expect a bit more reasons / explanations about why a group of algorithms fails in the studied setting. For example, in the second paragraph of Sec 4.1, we only get vanilla KD is better, but still do not understand why those complex methods just offer limited improvement.

3. Some typos:
can be reformulate as -> can be reformulated as \
the research question is prompt: … -> the research question is: … \
a evaluation framework -> an evaluation framework \

**Questions:**

See above.